# In Situ Observations on the Crack Morphology in the Ancient Timber Beams

Nicola Ruggieri 

Superintendence of Archeology, Fine Arts and Landscape for the Province of Cosenza, 87100 Cosenza, Italy; Nicola.ruggieri@unical.it

**Abstract:** The conservation of ancient structures is, in the construction panorama, a highly eco-sustainable operation. In fact, it provides for a very limited consumption of resources. This article provides an in-depth analysis of ancient wooden material, an essential element for drawing up correct conservation interventions. Ancient timber beams have a peculiar morphology of failure dependent on many factors, among which are the species of wood, the quality of the material-knots, presence of fissures caused by shrinkage (checks), direction of the grain, and environmental conditions such as temperature and humidity. In addition, it is linked to load conditions and static configuration. This paper presents a case study of failed ancient timber members still in place and describes the type of failure as well as the origin and propagation of the cracks. The objective is to provide a classification of the causes and of the effects and their evolution, useful to practitioners and to those who have to make decisions on the timber structures conservation.

**Keywords:** ancient timber beam; flexural failure; cracks morphology; diagnosis



## 1. Introduction

In 2016, Prof. Tampone, "tutelary deity" of the ancient timber carpentries, inaugurated the foundation of a science concerning the failures of the timber structures [1].

The proposed study presents a catalogue raisonné of different collapse expressions of ancient timber beams, in the wake of the recent studies conducted by the late lamented Professor. For that purpose, the analysis of some broken, still in situ, beams examined herein has allowed one to classify the rupture effects and their evolution.

The timber members under study are presented, when possible, in their historical scope and structural system context as well as in the loading regime in which they operate. Furthermore, the contribution provides data on the quality of the wood, strategic in the structural response. The damage is documented, in the eleven case studies, in its effects and to each one has been attributed its own specificity, speculating failure reasons from natural imperfections in the structural wood, to fire and earthquake damage exacerbated by construction methods. In this way, a systematic overview is drawn up of the causes, manifestation, and progression of damage, predominantly based on direct observation. The laboratory can reproduce only some of the conditions that characterize an ancient wooden beam in place. In fact, the duration of the test cannot take into account the complexity of the parameters that influence the behavior of a wooden beam for hundreds of years (i.e., the regime and progression of loads). Therefore, the importance of the presented research mainly lies in the fact that timbers are analyzed in situ. Such a condition is an inexhaustible source of learning on the structural behavior of wooden beams.

Moreover, the review of the recent literature regarding bending tests carried out on old timber beams in structural dimension has led one to make a hypothesis on the mechanical properties variation due to wood defects.

The number of researchers who have focused their studies on ancient carpentry is modest when compared to the number of scholars interested in new timber structures. The scientific literature that has dealt with the mechanical behavior of ancient beams has focused

its interest on defining the resistance property values [2–18]; brief notes on the proposed theme, conversely, have either been included in Piazza et al. [19] or indirectly deduced from the strength tests conducted on timber members from demolished buildings [4–6,8,20].

Therefore, the scientific literature does not present specific studies on the morphology of the fracture of beams in situ.

Tampone, in [1], presents few cases of beam failures observed during his long scientific and professional career. In fact, ancient beam breakage is rarely found. Replacement or restoration of floors often hides evidence of fracture and can thus prevent information being gained. It is only because the author has carried out numerous monument surveys in his role as an official of the Italian ministry of cultural heritage that the majority of the presented cases have been discovered.

## 2. A Complex Problem: The Variables

Timber structural performances are the result of the interaction of the genetic potential of the tree with the environment in which it grows. Wood is ascribed to a special category among building materials due to its peculiarities in the response to stresses. In fact, it highlights, in addition to the notorious anisotropy, an extreme heterogeneous behavior mainly in relation to the species of the wood; quality of the material, i.e., knots, presence of shrinkage checks, direction of the grain, etc.; and environmental conditions, i.e., temperature and humidity.

Moreover, the performances of the timber members are influenced, not only by the static configuration with which they are assembled, but also by the loading regime, as related to their time span and intensity [21,22]. This complexity is completely found in the morphology and progression of the rupture, and takes on peculiar aspects generating a multiform scenario. The latter is further complicated if the beams are characterized by the simultaneous presence of two or more of the factors listed above.

## 3. Materials and Methods

This paper provides a summary of common cracking patterns and failure mechanisms of ancient timber beams identified in multiple historic case study projects, ranging from the 16th to the early 20th century, and variable for load conditions and wood quality. The herein study includes, by means of a visual analysis (according to UNI 11119, 2004), the material characterization with particular regard to the defects and their position with respect to the load and a description of the timber members configuration.

The detected failure modes of timber beams—grouped in the following paragraphs as brittle/quasi brittle type, fatigue phenomena, and pseudo-ductile type—have been identified basing on in situ investigation (i.e., convent of San Daniele in Belvedere Marittimo; factory in the province of Cosenza; Palazzo Tassoni Estense in Ferrara; Sanctuary of the "Madonna della Grotta" in Praia a Mare; others on photos observations (i.e., house in Marina di Pietrasanta (Lu); church of Notre Dame in Paris; the church of the San Fernando Rey de España; Royal Palace in Naples; beams subject to bending test).

The well-known collapse mechanism for bent timber elements of "pseudo-ductile" type, theorized during the seventies and eighties [23–26], is determined by the difference between the characteristic value of the ultimate resistance to tension and that to compression stress. Therefore, as the stress level increases in a beam, a first plasticization phase appears at the upper edge of the beam that manifests itself with corrugation and grain separation due to the instability—buckling—of the compressed fibers. Such a situation activates the shifting toward the lower side of the neutral axis and tensile failure with a saw-toothed tearing of the fiber bundles and rupture lines perpendicular to the edge. This collapse modality finds evidence in the investigated ancient structures, although early tensile brittle crisis triggered by wood "defects", that precede the mobilization of deformations at the upper side, are more recurrent.

Moreover, the analysis of the recent literature concerning experiments on salvaged beams has made possible to evaluate the variability of the properties of bending stiffness

and strength of aged timbers correlated to the defect that triggers the rupture. In other words, regardless of the load history, however an important parameter, it has been possible to establish how much, knots, slope grain, checks, and internal anomalies are able to affect the modulus of elasticity (MOE) and the modulus of rupture (MOR) values.

The experimental data come from four-point bending tests [3–5,8,9] according to the UNI EN 408 1995 and from three-point flexural tests relying on ASTM standards [2].

## 4. Results and Discussion

### 4.1. Rupture Cases of Brittle/Quasi-Brittle Type

Saint Daniele's convent in Belvedere Marittimo (Cosenza, Italy), whose first structure was built by the Capuchin Friars Minor and dates back to the end of the 16th century, is characterized by inter-story floors with one order of beams and timber boards above.

Some of the beams were likely shaped by using an adze, from which it can be inferred that the surfacing of the log was not recent. The used wood specie, determined by the macroscopic features of the material, is Castanea sativa Mill. The beams have, in general, a scarce wood quality, as they are affected by knots, even large, and by the grain deviation (Figure 1).

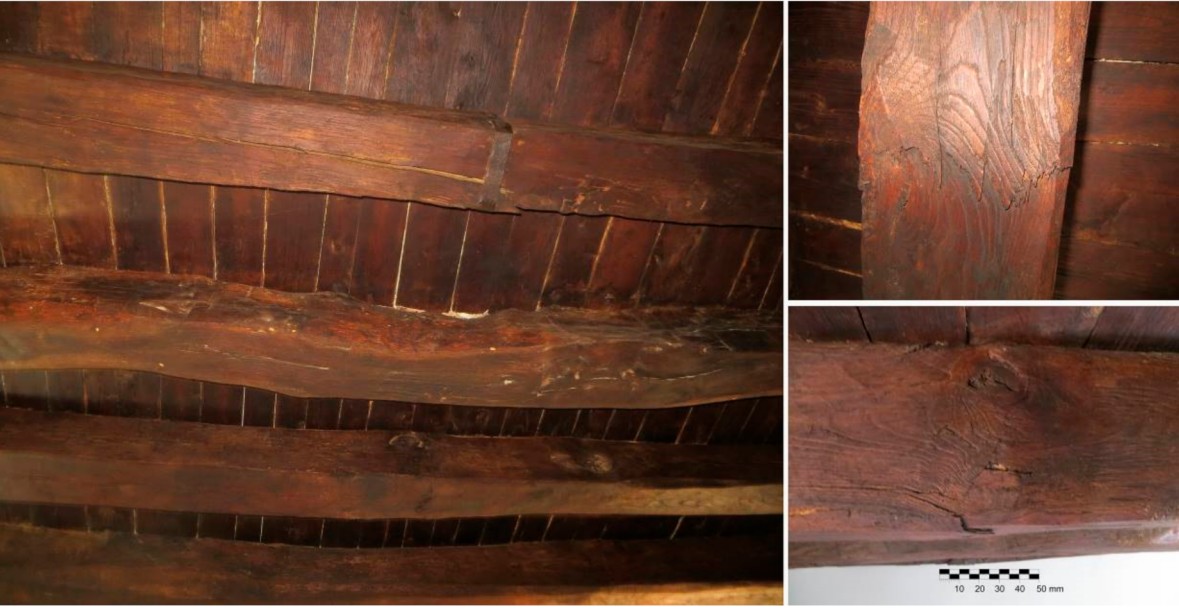

**Figure 1.** On the left: Timber floor of a room contiguous to the cloister of Saint Daniel's convent (Belvedere, South Italy). The beam in the foreground highlights fiber delamination facilitated by the shrinkage check and by the deviation of the grain. On the right above: The superficial crack at the lower edge of a beam of the Saint Daniel's convent triggered by a knot as well as by the local grain deviation. On the right below: The superficial crack has affected the face of a beam of the Saint Daniel's convent triggered by a knot as well as by the local grain deviation.

Furthermore, the arrangement of the resistant section does not comply with the possible maximum inertia deriving from the timber squaring—the width is greater than the height—even if such geometry provides an ampler support to the boards. Insect holes are diffusely present pointing out a biotic attack, perhaps extinct.

The room under study, located in the western part of the cloister, emphasizes two broken beams, yet despite this the entire floor collapse has not taken place in virtue of the structural cooperation characterizing, generally speaking, timber carpentry. The superficial crack that shows the fifth member (counting the beams from the north to the south side of the room), located approximately in the midpoint of the lower edge, is derived from possible manufacturing imperfections. It has been triggered by the combined effects of the material discontinuity caused by a knot and by the local grain deviation, i.e., a physiologic

consequence of the tree adaptation to the variation originated by the branch. The knot involved a modification of the stresses path with a concentration of the stress value near the defect and punctual overcoming of the resistant capacity of the timber. The consequent rupture occurred with the tissue laceration that follows a circumferential path to the knot. The fibrousness of the borders is more evident as the fracture surface becomes more distant from the "defect", where the peculiar pattern of the crack is mainly governed by the local irregularity of the grain (see Figure 1). Then, the fracture is propagated in multiple branches, one of which is characterized by a subhorizontal trend that intercepts a check, i.e., a discontinuity of the material that occurs in wood due to the effect of the anisotropy of the shrinkage, in the middle part of the face.

The other branch of the crack is "attracted" by a further knot localized near the compressed side (see Figure 1). In that case, the borders show moderately jagged surfaces, almost flat, typical of an elasto-fragile behavior and of a volume subjected to compression stresses [27].

As a broad generalization, for particular loading conditions the shrinkage checks can turn into cracks, even if the reduction of the resistance of a beam owing to such a "defect" can be negligible [28]. Such a circumstance characterizes a beam part of the same timber context of the convent of San Daniele in Belvedere Marittimo, where the check exhibited at the lower edge with the complicity of the irregular arrangement of the fibres was a preferential way for the grain delaminating determined with sharp borders without fraying. An attempt has been made to remedy the partial grain separation by affixing, to restore the cooperation to the disjointed fibers, an iron cramp. The latter, basing on the conception, form, and workmanship that distinguish it, probably dates back to the end of the nineteenth and the beginning of the last century.

Similar crack morphology can be detected in a beam belonging to an inclined pitch roof of an early 20th century factory in the province of Cosenza, Italy. The presence of the top of the tree on the right side of the photo and therefore of a tapering of the resistant section led to a decentralization of the maximum deflection. The rupture occurred with delamination of the fibers, a typical morphology for cracks triggered by a deviation of the grain slope with respect to the longitudinal axis of the trunk (Figure 2).

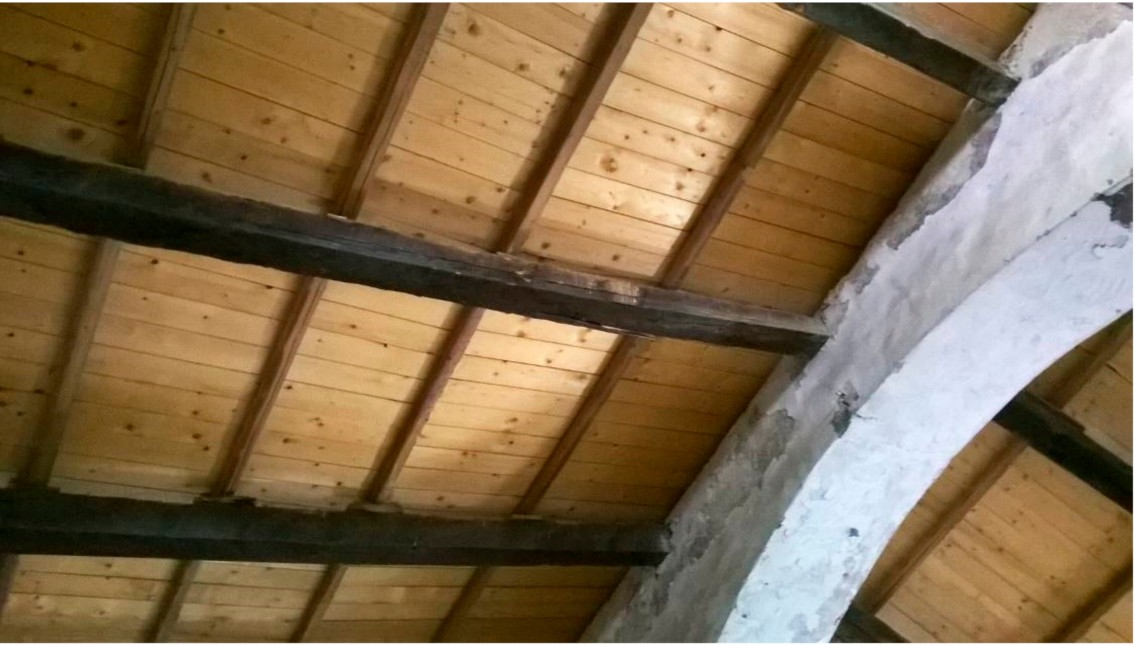

**Figure 2.** Cosenza (Italy), fiber delamination in a beam of the early 20th century.

The beam belonging to the roof carpentry of the Palazzo Tassoni Estense in Ferrara (Italy) is characterized by another type of anomaly with consequences for the failure morphology.

The building, now the headquarters of the Faculty of Architecture, was built in the first half of the fifteenth century. At the end of the fifteenth century, the building underwent an important restoration by Biagio Rossetti that probably involved the roof carpentry of the Hall of Honor. This structural system consists of two central timber trusses that, in addition to receiving the roof loads, had the extra burden of supporting the heavy chandelier that characterized the underlying salon (see Figure 3).

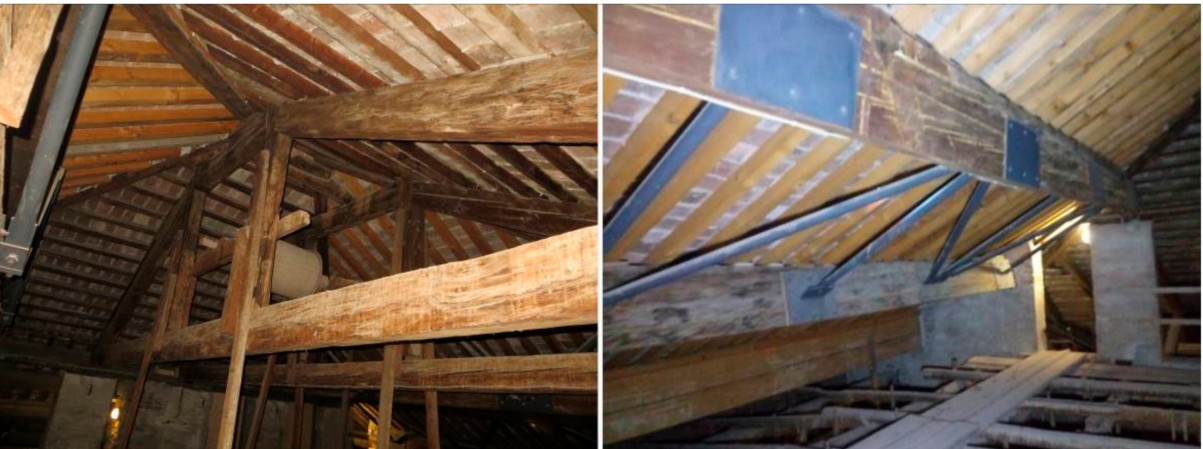

**Figure 3.** On the left: The two trusses of the roof carpentry of Palazzo Tassoni Estense in Ferrara. On the right: The structural units at the two lateral ends.

Additional supports for the covering are constituted by structural units at the two lateral ends recently subjected—in 1997—to reinforcement by inclined metal rods that have hybridized the structural element realizing lattice-type beams (see Figure 3). The horizontal members of the latter consist of two overlapping beams whose cooperation is obtained by means of quadrangular head nails. Therefore, holes are made that, in addition to interrupting the continuity of the grain, undermine the efficiency of the wood in the stresses transferring and distributing. In other words, a weakened section that exceeded the strength value of the beam entails the inevitable consequence of the crack triggering precisely along the solution of continuity. The resulting fracture under tensions is perpendicular to the longitudinal axis and has the typical jagged tips, although the length of the fiber bundle tears is modest. The crack progressed by affecting the faces of the beam where it assumed a sloping development that enclosed the "disturbance" element constituted by a knot (see Figure 4).

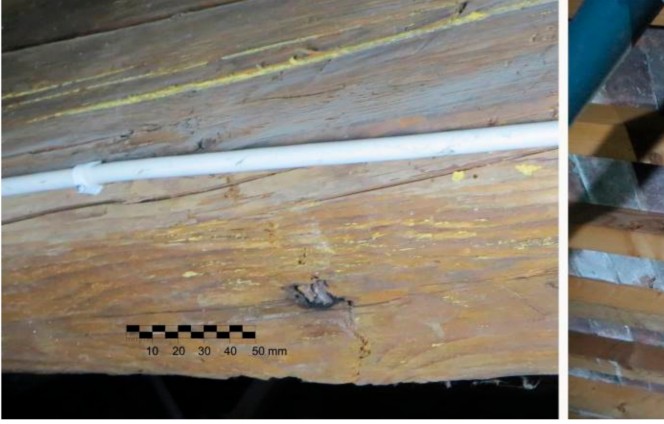
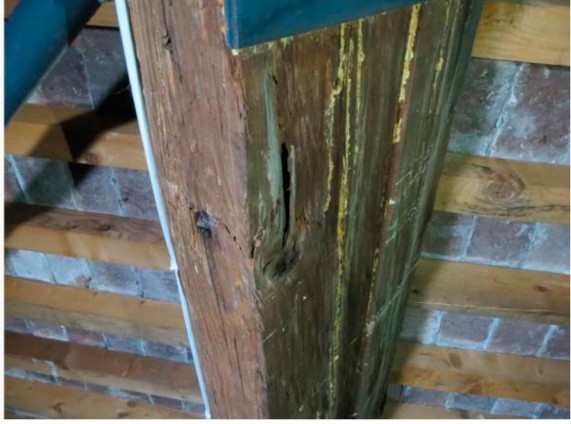

**Figure 4.** Roof carpentry of Palazzo Tassoni Estense in Ferrara. On the left: The crack was propitiated by the presence of a hole. On the right: The crack assumes a trend following the knot perimeter on the beam face.

The detectable failure on a beam in the aftermath of the terrible fire of 15 April 2019 that struck the church of Notre Dame in Paris is peculiar (Figure 5).

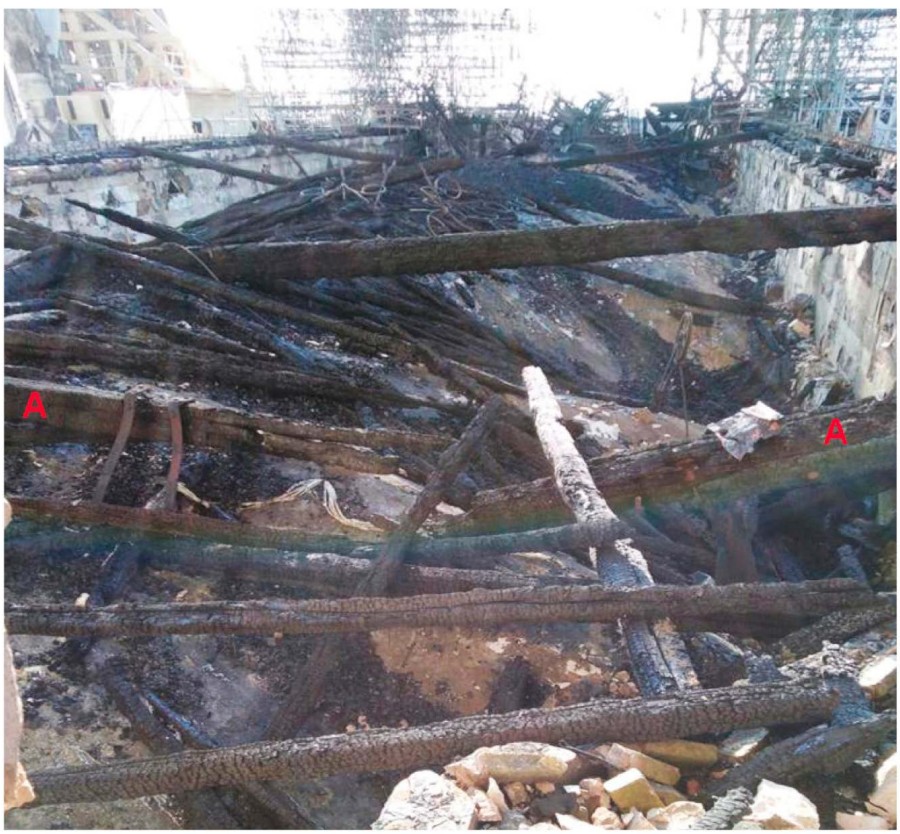

**Figure 5.** The beam (approximately in the middle of the photo with two iron strips, indicating by the letter "A"), belonging to the structural system of the roof of Notre Dame's in Paris, was broken in two parts after the fire of the 15 April 2019.

The member (indicated by the letter "A" in the Figure 5) appeared after the fire divided into two parts with a rupture occurring approximately in the middle. It can be assumed that the collapse was propitiated by a shrinkage check. In fact, it is a plausible hypothesis that the latter created a discontinuity in the insulation of the inner part of the wood, increasing its exposed surface to the fire. Thus, the carbonization entailed a start of splitting into the check with progressive divarication of the two extremities until breaking into two stumps due to the partialization of the resistant section. The regular course of the described solution of continuity along subparallel axes to the longitudinal axis of the beam leads one to presume a regular and straight grain.

The ribs of the vaulted lathwork can be ascribed among the beam typologies, whose curved shape implies a cut of the fibers along an inclined plane as respects to the applied force [1]. This is a predisposing factor for the splitting of fiber bundles, with a fracture morphology similar to what occurs in the case of a beam with cross-grain wood. Such a failure modality was found in the low profile vaulted ceiling of the 19th century Sanctuary of the "Madonna della Grotta" in Praia a Mare, Cosenza, Italy (Figure 6).

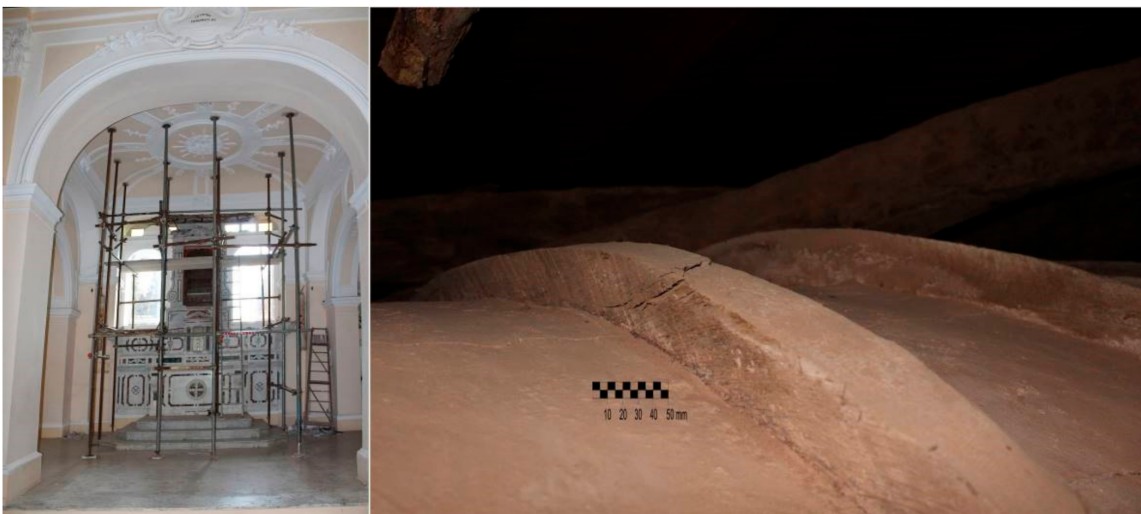

**Figure 6.** On the left: The intrados of the Santa Maria della Grotta's vault (Praia a Mare, South Italy). On the right: The negative bending moment in the rib, near the springer, has generated cracks parallel to the grain.

The carpentry, deriving from the Philibert de l'Orme's system, is constructed of a lathing supported by ribs of one piece of approximately 10 cm in height that covers a span of 5 mt, with the aid of the beams above. The excessive deformation of the latter and the consequent lack of contribution to the strength and the stiffness of the structural system have caused a flexural rupture.

Such damage has induced a negative bending moment near the springer, whose tensile stresses acting on the extrados have generated cracks parallel to the grain (see Figure 6).

Another case of fragile rupture can be recognized in a carpentry belonging to a house in Marina di Pietrasanta (Lu), Italy. The crack started on the side subjected to tensile stresses is characterized by flat borders. Ultimately, the crack is propagated in the perpendicular direction along a shrinkage check further detaching the fibers from each other. The cause of the described failure is to be found in possible anomalies inside the wood (Figure 7), such as the separation of the fibers along the growth rings tangential direction or, in general, decay.

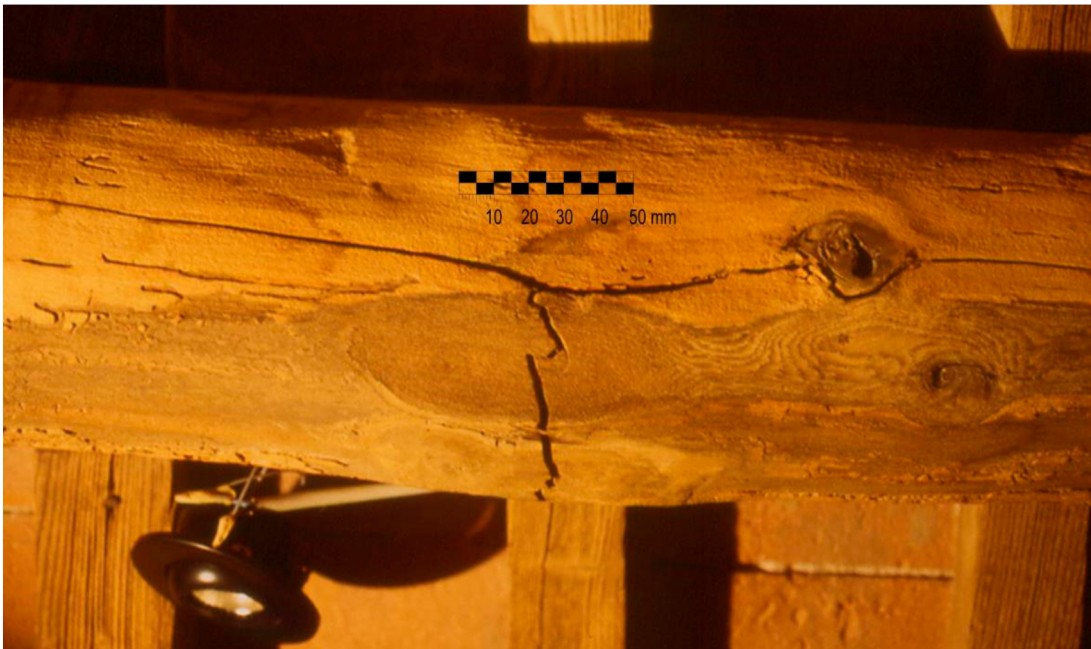

**Figure 7.** Failure due to wood internal anomalies in a beam of a house in Pietrasanta (Italy) (Photo Tampone G.).

*4.2. Fatigue Phenomena*

A wooden "text" particularly interesting for the variety of the failures that occurred is the roof of the church of the San Fernando Rey de España mission in California which dates back to the early 19th century [29]. The structural system is composed of couple close roof, with the addition of a king-post. Such members are characterized by a strong dimensional hierarchy, as far as can be ascertained from the late nineteenth century photo of the roof. In fact, the tie-beams have a section at least twice as high as the rafters. Further, there are cantilevers finely decorated at both the masonry supports, useful for increasing the resistant section near the maximum value shear stress of the beam and, at the same time, for reducing the span of the tie-beam as well as aimed at distributing the compression stress on a surface ampler rather than the underlying masonry.

The 1812 San Juan Capistrano earthquake, which registered an estimated magnitude 7 on the Richter Scale [30], caused diverse damage to the carpentry. Among which seems to be the collapse of some trusses, as far as can be deduced from the considerable inter-axes between some of the structural units and from the heaps of material near the group of people (Figure 8).

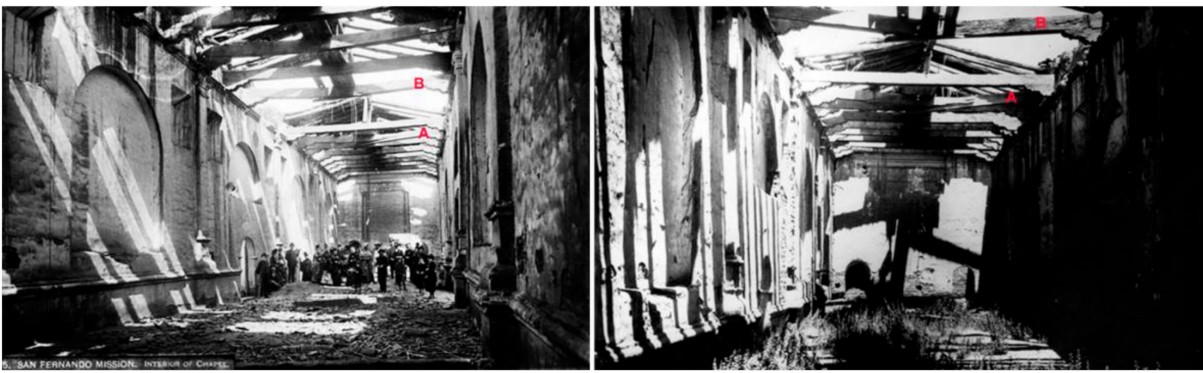

**Figure 8.** On the left: Photograph of the 19th century depicting the roof carpentry of the church of the San Fernando Rey de España in California struck by the 1812 earthquake. On the right: Early 20th century photograph of the church of the San Fernando Rey de España taken after those of the photo on the left (note the weed vegetation). The splitting between the fibers increased due to creep phenomena, favored by the exposure to high values of humidity, as well as to possible weakening of the resistant section caused by biodeterioration.

The repeated action of the nineteenth century quake has probably caused phenomena of oligocyclic fatigue on the wooden structures [13] with multiple connotations of failure in relation to the wood quality. In other words, a low number of cycles with inelastic response of the truss due to plastic deformation developed in the joints [31,32] may have imposed the damage below detailed. The hypothesized mechanism is the increasing of concentrated stresses by means of the king-post in the middle of the chord caused by the vertical component of the earthquake, in addition to the probable inadequate behavior of the tie beam-rafter joint that ensures the onset of tension stresses in the horizontal member. Ultimately, the repetition of stress has led to a significant deflection of the tie-beams presumably increased by the deformation contribution in the vertical plane, consequent to the Euler's critical load reaching due to a possible asynchronous oscillation of the church parallel walls.

The cracks found seem to have sharp borders, without evident jaggedness, with delamination between the fibers owing to the onset of tangential stresses. Such a phenomenology of failure in some cases is facilitated by the oblique cut of the grain deriving from considerable inclination of the fibers with respect to the axis of the development of the stem. The chord in the foreground of Figure 8 (indicated by the letter "B"), noticeably inflected, suffered splitting of fibers in two opposite directions near the center line. In addition, longitudinal fractures which extend, approximately, up to the neutral axis of

the section can be observed. The cracks can be recognized by the anomalous peak of the deformed shape of the bent member. Further turning into separate layers of the grain are found in the horizontal member of the truss highlighted in Figure 8 (indicated by the letter "A"), in correspondence with the king-post, with propagation of the fracture characterized by an inclined progression, in its linearity disturbed, probably, by the presence of a knot.

Cases of fractures parallel to the grain, more or less marked, caused by the longitudinal sliding of fiber bundles with, perhaps, sharp tips, can be observed in Figure 8.

### 4.3. Rupture Cases of Pseudo-Ductile Type

The organization of the roof carpentry of the Royal Palace in Naples is part of the interventions carried out in the 18th century by order of the King of Borbone to adapt the sixteenth century building to an archaeological museum, a place dedicated to the preservation of the significant quantity of finds coming from the Vesuvian cities [33–35].

The load-bearing structure of the roof, with a span of approximately 25 m, is composed of two types of trusses: One of which has the role of supporting the covering, the second, with a considerably reduced slope and therefore more rigid in the vertical plane, has the task of supporting the timber lathwork vault of the Central Hall ceiling (Figure 9).

The rafter is aided in its task by an under-rafter for both types of trusses. Furthermore, the main structural unit, belonging to the roof carpentry, is equipped with a tie-beam, two false ties (upper and lower), two king posts and struts that constitute intermediate supports of the long composite rafter. The latter is constructed by means of two pitch-pine (*Pinus rigida*) logs, just debarked, without squaring that can sever the continuity of the fibers.

The upper edge of one of the composite member-the under-rafter in particular-shows a nonlinear behavior with local instability of the compressed fibers and separation between them, an eloquent manifestation of the achievement of the resistance limit; there is no sign of damage in the volume stressed by tension (Figure 9).

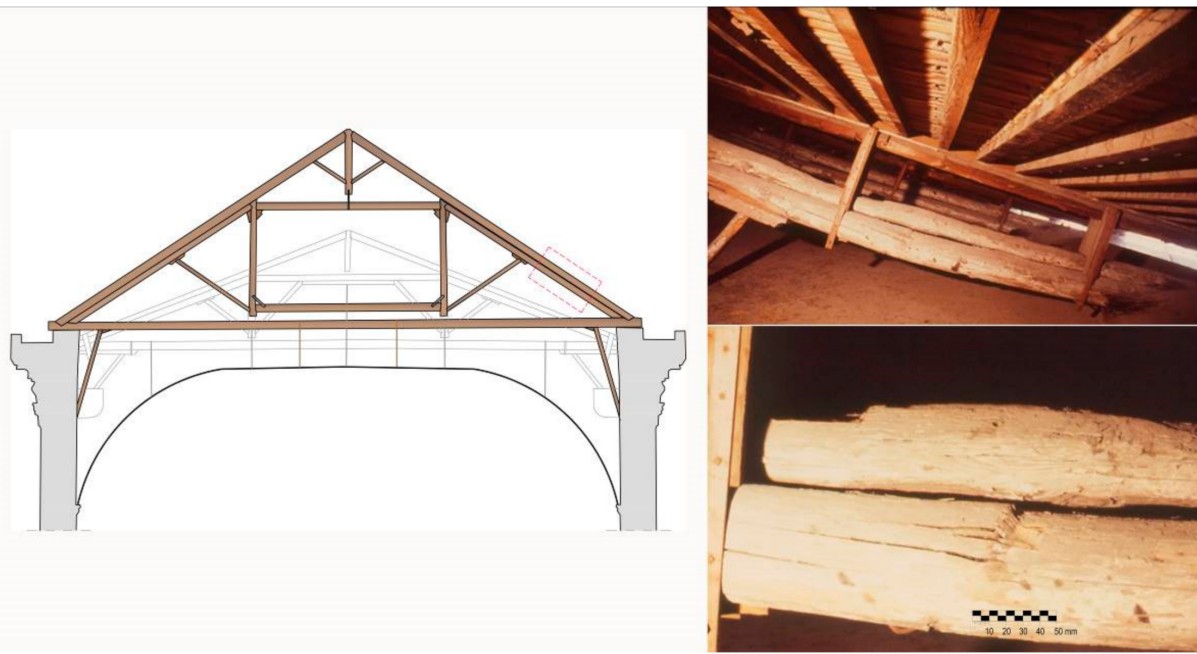

**Figure 9.** On the left: Roof load-bearing structure of the Royal Palace in Naples. There are two typologies of trusses. On the right above: Rafter and under-rafter of one of the Royal Palace trusses. The lower member is considerably deformed and shows the crushing-splitting fibers (Photo Tampone G.). On the right below: Roof trusses of the Royal Palace in Naples. Nonlinear behavior of the under-rafter with crushing-splitting of the fibers at the upper edge of the member (Photo Tampone G.).

Similarly, the beam taken from a disused building in the historical center of Naples [5] is only debarked and not squared. This specimen was used in the mechanical characterization tests conducted by a research group from the University of Naples Federico II.

The beam, subjected to a four points bending test until collapse, showed rupture morphology similar to that shown by the under-rafter in the Naples Archaeological Museum. In fact, the compressed volume has presented a local crushing of the grain attributed to the achievement of the ultimate compression strength value with, later, subsequent tensile collapse (Figure 10), according to the flexural failure theoretical model.

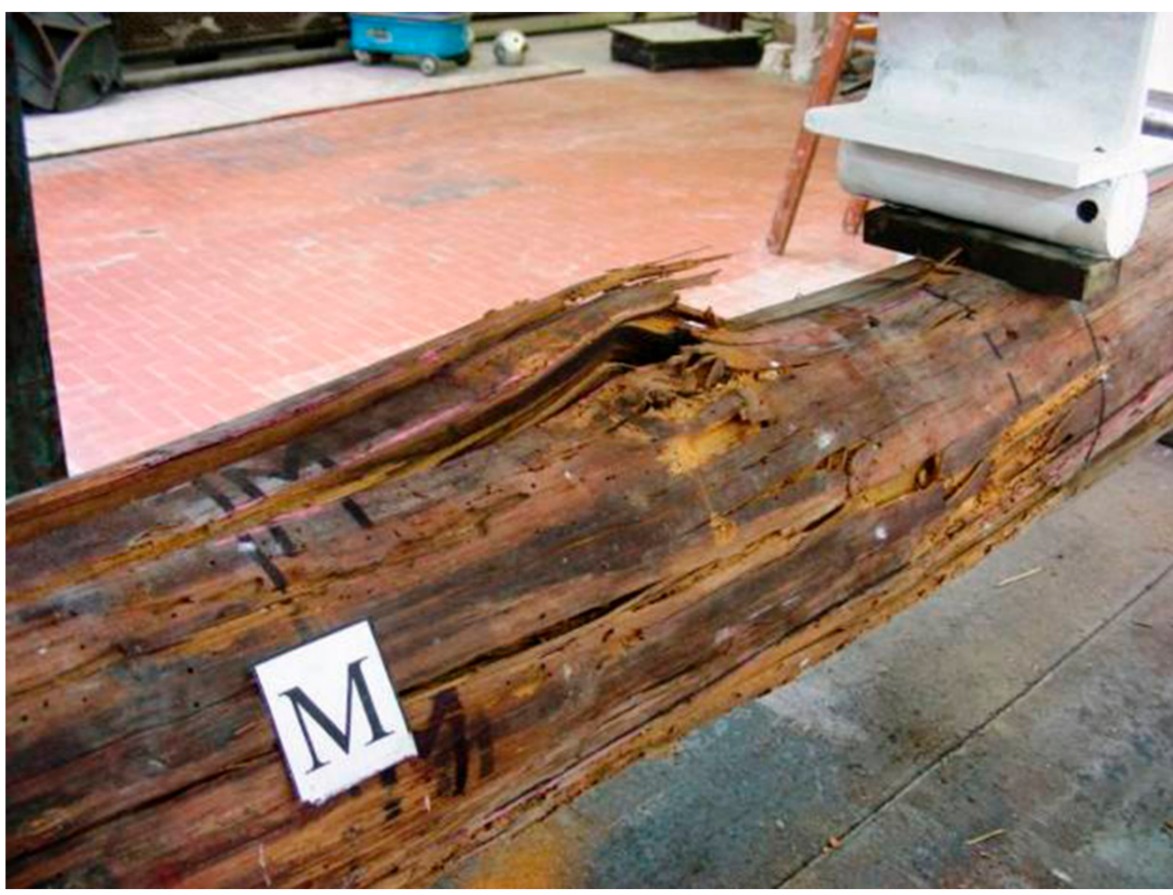

**Figure 10.** The tested beam shows the plasticization of the compressed fibers at the upper edge (Photo Faggiano B.).

### 4.4. Effect of Cracking Morphology on the MOR and MOE Values

High heterogeneity in the MOR results—governed by the strength of the "weakest" particle—and in the stiffness characteristics of aged members is highlighted by the analyzed experimental campaigns.

The MOE and MOR variation between old and new timber is summarized in the Table 1 and in the Schemes 1 and 2.

**Table 1.** Aged wood mechanical properties from the literature review.

| Wood Specie | Age | Specimen Number | Visual Grade | MOR (MPa) | MOE (GPa) | Rupture | Experimental Test Authors |
|---|---|---|---|---|---|---|---|
| *Pinus strobus* L. | 141 years old | 1 | Internal anomalies | 2.28 | 4.80 | Brittle | Attar-Hassan, 1976 |
| *Abies alba* Mill. | 15th C | 3 | Ring shake, checks | 27.16 (mean) | 10.03 (mean) | Brittle | Ceccotti et Togni, 1996 |
| *Abies alba* Mill. | 15th C | 3 | checks | 35.5 (mean) | 12.62 (mean) | Pseudo-ductile | Ceccotti et Togni, 1996 |
| *Abies alba* Mill. | 15th C | 3 | knots | 38.06 (mean) | 10.91 (mean) | Quasi-brittle | Ceccotti et Togni, 1996 |
| *Eucalyptus globulus* Labill | 1914 | 2 | Shrinkage checks | 83.4 (mean) | 19.65 (mean) | Brittle | Branco et al., 2005 |
| *Eucalyptus globulus* Labill. | 1914 | 1 | Slope grain | 47.2 | 17.4 | Brittle | Branco et al., 2005 |
| *Eucalyptus globulus* Labill. | 1914 | 1 | Clean wood | 98.5 | 16.5 | Quasi-brittle | Branco et al., 2005 |
| *Castanea sativa* Mill. | 19th C | 9 | Knot; not squared member | 40.44 (mean) | 12.37 (mean) | Quasi-brittle | Faggiano et al., 2010 |
| *Castanea sativa* Mill. | 19th C | 1 | not squared member | 46.91 | 15.33 | Pseudo-ductile | Faggiano et al., 2010 |
| *Castanea sativa* Mill. | Not specified | 1 | not squared member; III class | | 10.98 | | Branco et al., 2011 |
| *Castanea sativa* Mill. | Not specified | 5 | not squared member; II class | | 9.47 (mean) | | Branco et al., 2011 |
| *Castanea sativa* Mill. | Not specified | 1 | not squared member; I class | | 7.53 | | Branco et al., 2011 |
| *Abies alba* Mill. | Not specified | 3 | Grooves; slope grain | | 3.7 (mean) | | Cavalli et Togni, 2013 |
| *Abies alba* Mill. | Not specified | 3 | Grooves; knots | | 7 (mean) | | Cavalli et Togni, 2013 |
| *Abies alba* Mill. | Not specified | 3 | Grooves; shrinkage checks | | 3.8 (mean) | | Cavalli et Togni, 2013 |

During the bending tests performed in the Restoration Services Division lab in Ottawa, an unusual brash failure has emphasized a 141-year-old beam of eastern white pine characterized by internal anomalies [2] to which corresponded a MOR of 2.28 MPa, i.e., a reduction of 93% with respect to new wood.

The relevant presence of knots in silver fir beams belonging to a 15th century building and chestnut beams belonging to a 19th century building has, instead, emphasized average value of strength, respectively, of 38.06 MPa and 40.44 MPa, higher than new timbers MOR, respectively, +35% and +44% [3,10]. Furthermore, old silver fir timbers, in which a visual grading has pointed out shrinkage checks, have showed an average MOR value of 35.5 MPa (+7.5 Mpa with respect to new wood). The better performance has been exhibited by non-squared chestnut members—MOR equal to +18.91 Mpa, corresponding to +67%, if compared with strength property of fresh chestnut lumber, devoid of particular defects and to clean wood assimilated.

Shrinkage checks and slope grain are determinant wood defects in affecting strength properties of aged *Eucalyptus* members with values, respectively, of 62% and of 34% [4] lower than MOR value of new wood.

The time appears to affect also stiffness properties, although with lower percentages of variability and with more cases characterized by MOE values higher than new wood (Table 1). Old chestnut beams with significant dimension and wide distribution of knots as well as those graded as clean wood assumed values of the modulus of elasticity of 12.37 GPa and of 15.33 GPa, higher than the MOE of new wood, respectively, +12% and +39%; furthermore, experimental tests on old silver fir timbers have confirmed the good performance in terms of stiffness of aged timbers characterized by knots. On the other hand, the conditions that most affect the MOE are the internal anomalies—a value reduction of approximately 30% has been recorded—and the grooves coupled with different defects. The latter circumstance has led to MOE values lower than those of new wood and ranging between 60% less for members with deviation of the grain or checks and −30% for timber characterized by knots [9].

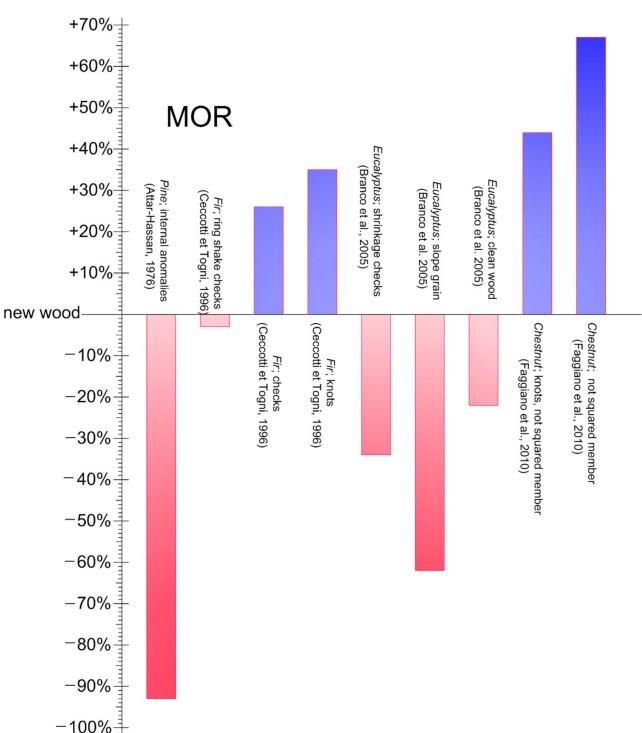

**Scheme 1.** Correlation between MOR values of aged and new timbers, from the literature review. Positive/negative values indicate higher/lower MOR than that of new timber. Source of new timber strength properties: chestnut, MOR 28 MPa; silver fir (category II), MOR 28 MPa (from "UNI, Ente Nazionale Italiano di Unificazione, 2009. Structural timber. Strength classes UNI EN 338:2009, Milano, Italy"); eucalyptus, MOR 127.5 MPa (from "LNEC–M6, 1997, Eucalipto comum., Fichas Técnicas, LNEC, Lisbona"); eastern white pine, MOR 34 MPa (from "Green, D. W. Winandy, J. E., Kretschmann, D. E. 1999, Mechanical properties of wood. Wood handbook: wood as an engineering material. Madison, WI: USDA Forest Service, Forest Products Laboratory. General technical report FPL").

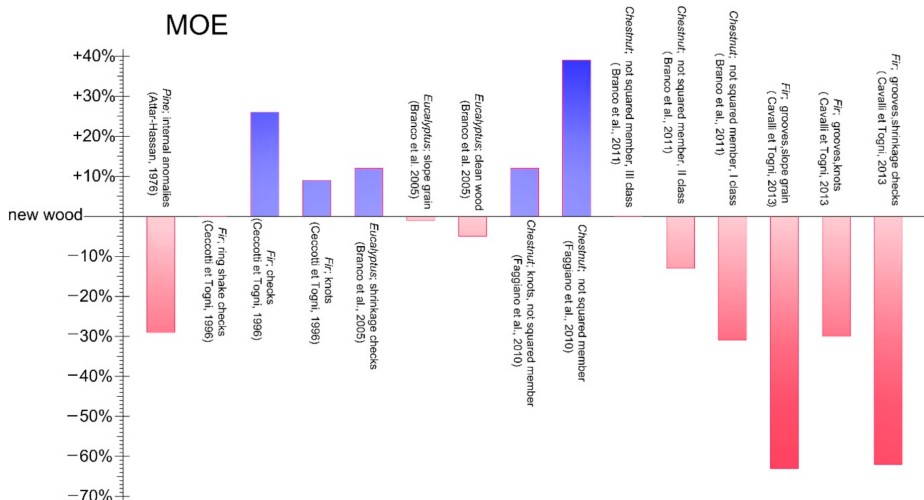

**Scheme 2.** Correlation between MOE values of aged and new timbers, from the literature review. Positive/negative values indicate higher/lower MOE than that of new timber. Source of new timber stiffness properties: chestnut, MOE 11 GPa; silver fir (category II), MOE 10 GPa (from "UNI, Ente Nazionale Italiano di Unificazione, 2009. Structural timber. Strength classes UNI EN 338:2009, Milano, Italy"); eucalyptus, MOE 17.5 GPa (from "LNEC–M6, 1997, Eucalipto comum., Fichas Técnicas, LNEC, Lisbona"); eastern white pine, MOE 6.8 GPa (from "Green, D. W. Winandy, J. E., Kretschmann, D. E. 1999, Mechanical properties of wood. Wood handbook: wood as an engineering material. Madison, WI: USDA Forest Service, Forest Products Laboratory. General technical report FPL").

The bending tests carried out by Branco et al. [4] on salvaged eucalyptus beams have provided a mean MOE value of 1830 GPa close to that of new wood.

A MOE decrease of about 30 percent is reported by Attar-Hassan [2] when compared new and aged wood of eastern white pine.

The stress–strain curve resulting from flexural tests is of extreme importance for the failure interpretation and progression (see Scheme 3). Chestnut beams with knots and without squaring, coming from a building in the historic center of Naples built in the nineteenth century, has showed a first linear-elastic branch up to the value of Fmax, from which has followed a slope variation and a branch with "steps", namely, a sudden loss of bearing capacity and a severe stiffness degradation.

A behavior definable as pseudo-ductile has been recorded in the experimental campaign on silver fir members with checks and silver fir members with knots—the latter's curve with no softening branch—performed in [3]. During such bending tests, the final load was preceded by an advancement of the crack, to which corresponds a series of "steps" in the scheme.

A similar shape has been assumed from the stress–strain scheme of a non-squared chestnut beam characterized by few knots [5]—identified as clean wood in the Scheme 3-which has emphasized a pseudo-horizontal trend following the achievement of its elastic limit; moreover, it has undergone large displacement in the plastic region without a moderate resistance impairment.

Conversely, the beam salvaged from a factory built in 1914 in S. Joao de Madeira, made of *Eucalyptus globulus labill.* wood [4] and characterized by deep checks that reach the heart of the tree, has showed a linear-elastic behavior with a brittle rupture immediately after reaching the Fmax value. The eucalyptus specimen graded as clean wood has exhibited the classical bending failure [4] with the plasticization of the compressed side and the successive crack generation at the lower edge of the beam. However, it has highlighted low ductility, if compared to the tested chestnut and silver fir beams.

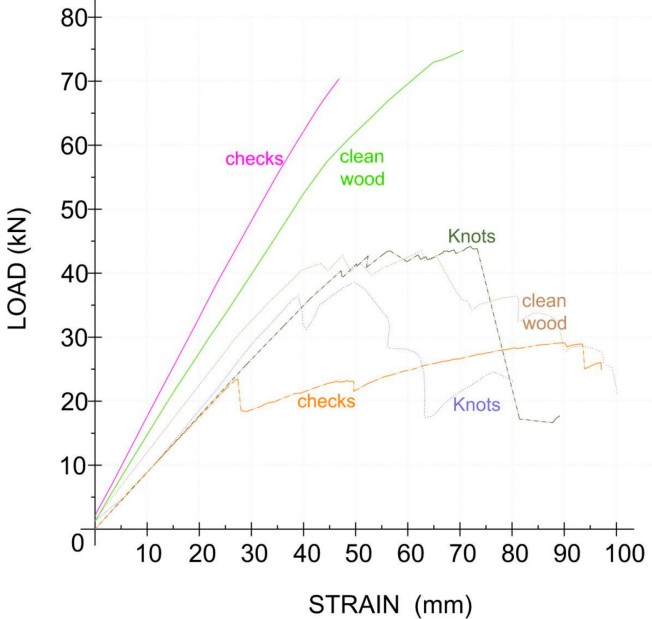

**Scheme 3.** Bending tests on old beam, stress–strain relationship, from the literature review. (Continuous line from "Branco, J., Cruz, P., Dias, S., (2005). Old Timber Beams Diagnosis and reinforcement, in proceedings of Conservation of Historic Timber Structures, Tampone ed., Firenze, 22–27 February, pp. 417–422"; Dash-dot line from "Ceccotti A., Togni M., (1996). NDT on Large Ancient Timber Beams: Assessment of the Strength/Stiffness Properties Combining Visual and Instrumental Methods, 10th International Symposium on Non-destructive testing on Wood, Lausanne, Switzerland, 26-27-28 August, Presses polytechniques et universitaires romandes"; Dotted line from "Faggiano, B., Grippa, M.R., Marzo A., Mazzolani F.M., (2010). Structural Grading of Old Chestnut Elements by Bending and Compression Tests. In: World Conference on Timber Structures. Riva del Garda (TN), 20–24 June 2010, p. ID-758").

### 4.5. Interpretative Hypotheses from the Crack Morphology and the Test Results Comparison

The presented survey of rupture cases of ancient wooden beams—although it does not consider a statistically significant number of samples and therefore requires caution in the discussion of the obtained data—highlights eloquent manifestations of collapse of timber on which to base some interpretative hypotheses. On this purpose different crack patterns linked with the main predisposing factors and the aspect of the fracture tips are synthetized in the Table 2.

**Table 2.** Main crack morphologies found in the analyzed ancient beams (* The pseudo-ductile Table 2. Main crack morphologies found in the analyzed ancient beams (* The pseudo-ductile behavior has been deduced from the literature on analyzed ancient beams bending tests).

| Main Predisposing Factor | Crack Pattern | Fracture Edges | Rupture Typology | Schema |
|---|---|---|---|---|
| Deviated Grain | Splitting and progressive divarication among the fibres | Sharp | Brittle |  |
| Knot | Circumferential to the knot shape | Moderately jagged/fibrous | Quasi-brittle/Pseudo-ductile * |  |
| Shrinkage Check | Divarication | Sharp/fibrous | Brittle/Pseudo-ductile * |  |
| Hole | Perpendicular to the longitudinal beam axis | Fibrous | Brittle |  |
| Internal anomalies | Crack perpendicular to the tree axis; Fibers divarication | Sharp | Brittle |  |
| Fibres Continuity (not squared beam) | Local crushing and fibres separation; corrugation | Sharp | Pseudo-ductile |  |

The presence of anomalies in the volume subject to tension of the beam in place is preferential ways for the crack triggering, namely, the distribution of stresses is governed by the inhomogeneity of the material.

Many of the described collapse cases were triggered by the grain deviation. The cause is to be found in the cutting, in an oblique direction with respect to the force, of the upset fibers; a condition that facilitates the establishment of cross grain tensions in the weak plane between the fibers, whose fragile transverse cohesion is ensured, at the cellular level, by the lignin. Moreover, it is worth noting that the tendency to fracture generation parallel to the grain finds more favorable conditions in ancient wood, rather than in a new one, due to the chemical processes linked to aging that lead to a degradation of the lignin [36,37]. A rupture morphology that corresponds to a substantial reduction in strength—for eucalyptus wood about −60% compared with new wood—while the incidence for MOE is contrasting [4,9]. In fact, despite 1% reduction of the stiffness value for eucalyptus wood, for silver fir wood characterized by grooves, beside slope grain, the MOE value is 63 percent lower than that of new wood. Furthermore, the stress–strain curve reveals a linear-elastic behavior with a brittle rupture.

The severe distortions of the tissues due to the knots that, in addition to representing a solution of continuity—the material is denser and therefore stiffer than the immediate surrounding wood—locally disturb the regularity of the grain and decisively influence the rupture which requires a lower fracture energy [38] compared to a defect-free timber beam. In this case, most predisposing factors are, as is known, the occurrence, size, quality and position of the knot with respect to the volume stressed by tension. However, experimental tests have revealed an excellent performance of silver fir and chestnut timbers in which knots are detected. In fact, the values of MOR and MOE are even higher than those referred to new wood [3,5]. The stress–strain curves of both wood species have exhibited a linear elastic behavior during a first stage and, after reached the yielding point, they have undergone plastic deformation characterized by localized ruptures. However, in the case of chestnut specimen a softening branch has been recorded, while fir beam curve a gradual moderate strength impairment has experienced, at least, until last load cycles. This contrasting behavior has to be attributed to the different size and distribution of knots in the two tested beams.

The peculiarity of the morphology of the tips of the fracture emerges from the analysis of the ruptures found in situ (see Table 2). In cases of rupture triggering caused by deviated grain they are flat—from which a brittle fracture is deduced [26,27,39]—but otherwise moderately fibrous, for example, if the resistant section is reduced due to the presence of holes. For the latter condition, a slight jagged fracture is observed, differently from the frayed edges with more elongated tears typical of cracks of the new defects-free wood [27,40]. Similar failure morphology is recorded in presence of anomalies inside the beam which, in addition to a significant deterioration of the mechanical properties, i.e., MOE $-93\%$, MOR about $-30\%$ if compared with new wood values [2], lead to cracks with flat tips.

If the fracture involves shrinkage checks the timber behavior is conflicting. According to the analyzed case study and the data of the Branco's lab tests [4] the crack is characterized by sharp borders, from which a fragile rupture can be deduced. Such a breakage typology has confirmed by the linear elastic trend assumed by the strain-stress curve (see Scheme 3). Instead, the experimental campaign on silver fir beams with checks that do not reach the heart of the tree carried out by [3] show excursions of the curve in the plastic region. The latter appears, therefore, as a pseudo-ductile failure due to the shallow discontinuity.

Cyclic bending loads due to seismic action induce damage to the beams similar to those attributable to the gravitational loads. In fact, at least for the examined example, longitudinal sliding along the deviated grain have been found culminating in splitting and divarication between the tensile fibers, ascribable essentially to elasto-fragile damage.

Ancient debarked and not squared beams are the specimens selected in which the pseudo-ductile rupture occurred thanks to the local plasticization arisen at the upper compressed side. The continuity of the fibers achieves a reduction of the unitary stresses thanks to the distribution of the tensile stresses on a greater number of fibers with respect to a not intact grain structure and, as a consequence, allowing the maximum utilization of the resistance to compression stresses of the section. The phenomenon finds a further probable motivation: the presence of the grain without solutions of continuity laterally confines the fibers stressed by compression, allowing for the perimeter ones to establish deformations deriving from the critical Eulerian load overcoming. Moreover, it can be assumed that the continuity of the fibers for the entire log section can inhibit premature tensile ruptures caused by cross grain in the wood of the beam. These considerations are confirmed by the results obtained by an experimental campaign carried out on chestnut members of the 19th century. In fact, those tests provided values of strength and stiffness, respectively equal to $+44\%$ and to $+12\%$, higher than the MOR and MOE values of new wood [5]. It is worth emphasizing that the latter two approximately circular cross section beams analyzed have highlighted a decidedly unusual mode of crack progression and, for that reason, need further investigation. In fact, the knot, exhibited at the lower edge of both beams, even if the typology is not determinable—if it is sound, unsound, encased, etc.—did

not impose an anticipated crisis at the tensile side, but the fibers crushing-splitting and the corrugation at the upper edge occurred, testing to the maximum the characteristics of the compression strength of the section.

## 5. Conclusions

The study offers a rational catalogue of cracks and failures with which it is possible to compare different symptoms that in professional practice can be found in a given structural context.

The evidence of the state of collapse, classified with the procedure proposed in the herein article, represents a contribution rare in the scientific literature to the slow and complicated knowledge of the mechanical behavior of the ancient wood material. This specialist contribution to the timber structures diagnosis science is essential to allow the interpretation of the state of crisis detected, guiding to the identification of the predominant cause, useful for being able to promptly prepare the necessary conservation measures.

The comparison between the mechanical values—in terms of MOE and MOR—relating to new and old wood has showed conflicting results with a great disparity between the results. In this regard, in order to fill this gap, in the near future the author intends to carry out an experimental campaign on ancient beams. The goal is to measure the response in relation to the load duration, in addition to the magnitude of the load. A dozen ancient beams, taken from buildings dating back to the sixteenth and nineteenth centuries, are available. These samples will be brought to rupture with different duration of the load (instantaneous, short-term, medium-term, long-term) from which to derive, in addition to the mechanical values, data on the progression of the damage by means of DIC investigation. The objective is to provide a contribution in the structural assessment and strength calculation of ancient carpentry. Furthermore, studies using targeted numerical methodologies are planned [34,35] to allow for a more rigorous interpretation of the observed in situ damage.

Moreover, the article corroborates some thesis of the recent theoretical debate around the importance of the semantic value of the degradations [41–43]. The deformations and the cracks, in fact, constitute attestations of the history of the construction and enrich the timber text. They are of extreme interest to retrace the events suffered and, therefore, add value to the intrinsic one possessed by ancient timber carpentry.

These considerations should be taken into account when preparing a conservation project.

**Funding:** This research received no external funding.

**Institutional Review Board Statement:** Not applicable, the study did not require ethical approval.

**Informed Consent Statement:** Informed consent was obtained from all subjects involved in the study.

**Data Availability Statement:** Data are not publicly available, though the data may be made available on request from the corresponding author.

**Acknowledgments:** The author is grateful to Beatrice Faggiano (University of Naples Federico II) and Jorge Branco (University of Minho) for sharing their experimental campaign results.

**Conflicts of Interest:** The author declares no conflict of interest.

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
