# Peer review of "In Situ Observations on the Crack Morphology in the Ancient Timber Beams"

_sustainability, doi:10.3390/su13010439_

Round 1
Reviewer 1 Report
An interesting article illustrating the problem of assessing old wood in existing structures:
It is not an easy question. So compliments for taking up this analysis. Therefore, below are some comments that may positively affect the article.
1) The readability of tables 1 and 2 should be improved. The tables are not legible.
2) There is no information on the number of trials or standard deviation. It is not known if these are random results or if there is any repeatability.
3) Lack of information about new wood. How was the quality of old wood determined and how was the mechanical properties of old wood determined to be able to compare it.
4) The article defines very general conclusions. It is worth specifying them.
5) There is a lack of how to take into account the condition of old wood in the structure in strength calculations.
A side note for the future:
Perhaps it is worth referring to the method of visual assessment of construction timber (almost every country has its own regulations) and proposing which of these criteria could be used to assess timber. It is also possible to indicate which of the visual assessment criteria can be omitted in the visual assessment and should be included in the computational analysis. The tangent structure of the structure is known for a specific element in the structure, so some defects (e.g. cracks, non-parallel course of fibers) can be included in the control calculations.
Author Response
Dear REVIEWER 1
Thank you very much for your efforts and time dedicated for evaluation of my manuscript. I do appreciate all your valuable comments and suggestions. Considering your overall assessment, please allow me to address your comments (in bold italic) point by point, as presented below.
An interesting article illustrating the problem of assessing old wood in existing structures:
It is not an easy question. So compliments for taking up this analysis. Therefore, below are some comments that may positively affect the article.
1) The readability of tables 1 and 2 should be improved. The tables are not legible.
The font size has been increased for better readability of the tables.
2) There is no information on the number of trials or standard deviation. It is not known if these are random results or if there is any repeatability.
The analyzed experimental campaigns provide very conflicting data. Another important factor in the analysis is that the trials available in literature is few. Therefore, it is not possible to define a statistical average or a standard deviation. In this regard, the author intends to carry out tests on ancient wooden beams of different wood species, to expand the results.
3) Lack of information about new wood. How was the quality of old wood determined and how was the mechanical properties of old wood determined to be able to compare it.
For new wood, the MOE and MOR values were drawn from the classification produced by “UNI, the Italian National Unification Body, 2009. Structural timber. Strength classes UNI EN 338: 2009”, considering category II and the same wood species of the ancient wood with which it is compared. The classification of the quality of the ancient wood was carried out by visual analysis. The mechanical properties of ancient wood derive from four-point bending tests according to the UNI EN 408 1995 and from three-point flexural tests relying on ASTM standards.
4) The article defines very general conclusions. It is worth specifying them.
The conclusions have been expanded by emphasizing the results obtained and giving emphasis to the degree of novelty. In addition, the studies that are planned by the author in the short future have been described.
5) There is a lack of how to take into account the condition of old wood in the structure in strength calculations.
It is a very interesting topic, in fact there are no studies on these issues in the scientific literature, nor is it addressed in the article. In this regard, in the near future it is the author's intention to carry out an experimental campaign on ancient beams with the aim of measuring the response in relation, as well as the extent of the load, to the duration. The experiments should consider a dozen ancient beams (already available) dating back to the sixteenth and nineteenth centuries. The trials will include different duration of the loads (instantaneous, short-term, medium-term, long-term) from which to derive mechanical values useful in the strength calculations of ancient carpentry.
A side note for the future:
Perhaps it is worth referring to the method of visual assessment of construction timber (almost every country has its own regulations) and proposing which of these criteria could be used to assess timber. It is also possible to indicate which of the visual assessment criteria can be omitted in the visual assessment and should be included in the computational analysis. The tangent structure of the structure is known for a specific element in the structure, so some defects (e.g. cracks, non-parallel course of fibers) can be included in the control calculations.
In the specific case the reference was the Italian UNI 11119 (Beni culturali - Manufatti lignei - Strutture portanti degli edifici - Ispezione in situ per la diagnosi degli elementi in opera). A visual assessment designed specifically for the analysis of cracks could be very original. In this regard, the use of algorithms with instrumental analyzes could also be interesting, which in addition to the defect, highlight any cracking pattern.
Reviewer 2 Report
The paper presents an interesting and original research. However several remarks, suggestions and recomendations are provided:
Introduction: it is suggested to include more literateure reference based on previous research, as well as make more enphasis in the relevance of the research, as well as in the knowledge gaps that it is adressing.
Method and Results: it is quite strange the organization of these two parts. Because the description of the method includes part of the obtained results. It is suggested to provide a clearer description of the methods and scope of the study. Also, the results can be considered as the discussion of results. Thus, it is recommended to reconsider its organization.
Conclusion: it is recommended to make more enphasis in the findings an novelity of the results. As well as provide more details about the future research.
Author Response
Dear REVIEWER 2
Thank you very much for your efforts and time dedicated for evaluation of my manuscript. I do appreciate all your valuable comments and suggestions. Considering your overall assessment, please allow me to address your comments (in bold italic) point by point, as presented below.
The paper presents an interesting and original research. However several remarks, suggestions and recomendations are provided:
Introduction: it is suggested to include more literateure reference based on previous research, as well as make more enphasis in the relevance of the research, as well as in the knowledge gaps that it is adressing.
Further literature references have been added:
Kuipers, J., Effect of Age and/or Load on Timber Strength, International Council for Building Research Studies and Documentation, Working Commission W18 – Timber Structure, September 1986, Florence.
Yokoyama, M.; Gril1,J.; Matsuo, M.; Yano, H.; Sugiyama, J.; Clair, B.; Kubodera, S.; Mistutani ,T.; Sakamoto, M.; Ozaki, H.; Imamura, M.; Kawai, S.. Mechanical characteristics of aged Hinoki wood from Japanese historical buildings, Comptes Rendus Physique 10, 7 (2009) 601-611.
Rug, W. ; Seemann, A. Strength of old timber, Build. Res. Inf. 19 (1991) 31–37.
Erhardt, D.; Mecklenburg, M.F.; Tumosa, C.S.; Olstad, T.M. New versus old wood: differences and similarities in physical, mechanical, and chemical properties, in: Int. Counc. Museums- Committee Conserv. 11th Trienn. Meet., London, UK, 1996: pp. 903–910.
Hirashima, Y.; Sugihara, M. ; Sasaki, Y.; Kosei, A.; Yamasaki, M. Strength properties of aged wood III: static and impact bending strength properties of aged keyaki and akamatsu woods, Mokuzai Gakkaishi. 51 (2005) 146–152.
Sousa, H.S.; Branco, J.M.; Lourenço, P.B. Characterization of cross-sections from old chestnut beams weakened by decay, Int. J. Archit. Herit. 8 (2014) 436–451.
Furthermore, it has been shown that the scientific literature on the subject has paid little attention to the dealt topic.
The importance of research mainly lies in the fact that wooden beams are analyzed in situ. In fact, the laboratory can reproduce only some of the conditions that characterize an ancient wooden beam in place. In fact, the duration of the test cannot take into account the complexity of the parameters that influence the behavior of a wooden beam for hundreds of years (i.e. the regime and progression of loads). Hence, the “in-place” condition is fundamental for an inexhaustible source of information.
Method and Results: it is quite strange the organization of these two parts. Because the description of the method includes part of the obtained results. It is suggested to provide a clearer description of the methods and scope of the study. Also, the results can be considered as the discussion of results. Thus, it is recommended to reconsider its organization.
The Method and Results paragraphs have been re-organized as suggested. Furthermore, the methods used and the purpose of the study have been better clarified.
Conclusion: it is recommended to make more enphasis in the findings an novelity of the results. As well as provide more details about the future research.
The conclusions have been extended by emphasizing the results obtained and has been provided more details on the future studies.
The comparison between the mechanical values ​​- in terms of MOE and MOR - relating to new and old wood has showed conflicting results with a great disparity between the results. In this regard, in order to fill this gap, in the near future the author intends to carry out an experimental campaign on ancient beams. The goal is to measure the response in relation to the load duration, in addition to the magnitude of the load. A dozen ancient beams, taken from buildings dating back to the sixteenth and nineteenth centuries, are available. These samples will be brought to rupture with different duration of the load (instantaneous, short-term, medium-term, long-term) from which to derive, in addition to the mechanical values, data on the progression of the damage by means of DIC investigation. The objective is to provide a contribution in the structural assessment and strength calculation of ancient carpentry. Furthermore, studies using targeted numerical methodologies are planned to allow for a more rigorous interpretation of the observed in situ damage.
Round 2
Reviewer 2 Report
The author has addressed to the provided suggestions and recommendations; however, the description of the methods and the research methodology can be better explained, in order to provide a clearer information to the reader. For example, describe the scope of the study (how many cases studies), and justify the statement about "the possible failure modes of timber beams" (why are those, and not less or more?).
Author Response
The author is grateful to the anonymous referee for the valuable comments.
All suggestions have been followed.
Additions to the text has been highlighted in green in the revised manuscript.
The author has addressed to the provided suggestions and recommendations; however, the description of the methods and the research methodology can be better explained, in order to provide a clearer information to the reader. For example, describe the scope of the study (how many cases studies), and justify the statement about "the possible failure modes of timber beams" (why are those, and not less or more?).
The methods and the research methodology has been better explained:
“The under study timber members are presented, when possible, in their historical scope and structural system context as well as in the loading regime in which they operate. Furthermore, the contribution provides data on the quality of the wood, strategic in the structural response. The damage is documented, in the eleven case studies, in its effects and to each one has been attributed its own specificity, speculating failure reasons - from natural imperfections in the structural wood to fire and earthquake damage exacerbated by construction methods -. In this way a systematic overview is drawn up of the causes, manifestation and progression of damage, predominantly based on direct observation. The laboratory can reproduce only some of the conditions that characterize an ancient wooden beam in place. In fact, the duration of the test cannot take into account the complexity of the parameters that influence the behavior of a wooden beam for hundreds of years (i.e. the regime and progression of loads). Therefore, the importance of the presented research mainly lies in the fact that timbers are analyzed in situ. Such a condition is an inexhaustible source of learning of the wooden beams structural behavior.”
The statement "the possible failure modes of timber beams" has been replaced with “the detected failure modes of timber beams”